# Health Care Monitoring and Treatment for Coronary Artery Diseases: Challenges and Issues

**DOI:** 10.3390/s20154303

**Published:** 2020-08-01

**Authors:** Mokhalad Alghrairi, Nasri Sulaiman, Saad Mutashar

**Affiliations:** 1Department of Electrical and Electronic Engineering, Faculty of Engineering Universiti Putra Malaysia; Universiti Putra Malaysia, Serdang 43400, Selangor, Malaysia; mokhalad.khalel@alkadhum-col.edu.iq; 2Department of Computer Techniques Engineering, Imam Al kadhum College (IKC), 10087 Baghdad, Iraq; 3Department of Electrical Engineering, University of Technology, 10066 Baghdad, Iraq; Saad.m.abbas@uotechnology.edu.iq

**Keywords:** in-stent restenosis, coronary artery disease (CAD), wireless pressure sensor, drug-eluting stent (DES), hyperthermia, computed tomography angiography (CTA), X-ray, radio frequency (RF) resonant heating, temperature regulation

## Abstract

In-stent restenosis concerning the coronary artery refers to the blood clotting-caused re-narrowing of the blocked section of the artery, which is opened using a stent. The failure rate for stents is in the range of 10% to 15%, where they do not remain open, thereby leading to about 40% of the patients with stent implantations requiring repeat procedure within one year, despite increased risk factors and the administration of expensive medicines. Hence, today stent restenosis is a significant cause of deaths globally. Monitoring and treatment matter a lot when it comes to early diagnosis and treatment. A review of the present stent monitoring technology as well as the practical treatment for addressing stent restenosis was conducted. The problems and challenges associated with current stent monitoring technology were illustrated, along with its typical applications. Brief suggestions were given and the progress of stent implants was discussed. It was revealed that prime requisites are needed to achieve good quality implanted stent devices in terms of their size, reliability, etc. This review would positively prompt researchers to augment their efforts towards the expansion of healthcare systems. Lastly, the challenges and concerns associated with nurturing a healthcare system were deliberated with meaningful evaluations.

## 1. Introduction

A coronary artery disease (CAD) is the gradual contraction of coronary arteries when plaque accumulates on the arterial walls. This hinders and decreases the flow of blood through the arteries, which might lead to a heart attack or stroke [1]. The arterial wall material that leads to narrowing is a waxy substance known as plaque. It is primarily made up of smooth muscle cells, macrophage cells, and complex extracellular materials such as sulphated glycosaminoglycan, fibrin, collagen, and cholesterol [2,3]. An invasive medical procedure called percutaneous coronary intervention (PCI) has become an important method to treat arterial plaques. This procedure makes use of the high-pressure expansion of a balloon in order to break the arterial plaque and enlarge the diameter of the vessel [4]. As demonstrated in Figure 1, percutaneous coronary intervention has been successful in treating patients and improving their symptoms. Furthermore, this procedure is more comfortable for patients since it is significantly less invasive than coronary artery heart bypass graft surgery. PCI is also relatively cheaper and leads to faster hospital recovery [5].

According to the World Health Organization (WHO), CAD heart and blood vessel diseases are responsible for around 30% of the total number of fatalities across the globe [8,9]. Since early detection of CADs results into easier management, continuous CAD monitoring has been increasingly necessary [10,11].

The American Heart Association estimates the direct and indirect costs and loss of work hours of CAD to touch USD1.1 trillion in 2035 [12]. Therefore, the utilization of Implantable Medical Devices IMDs, particularly stents, in relation to CAD is becoming an urgent and quickly evolving field.

The coronary stent refers to the metal scaffold that expands at the blockage site, forcing the artery open. Upon deployment, the stent is fixed using a balloon catheter. It is then fed to the blockage site using a guide wire. This is done via an incision in one of the femoral arteries (upper leg) of the patient. Upon correct location of the stent, the balloon catheter inflates in order for the stent to expand to its full diameter. The balloon catheter is then removed by deflating it [13].

The re-narrowing of the stent’s interior after its deployment is done to hold a blood vessel open, a process called in-stent restenosis (ISR) [14]. This re-narrowing is a result of the neointimal proliferation (scar tissue growth) within and around the stent. This happens as part of the immune response that the body has to the implantation of foreign objects [15]. Among stented patients, the possibility of in-stent restenosis can be as high as 50% [16,17]. It is therefore important to diagnose and treat in stent restenosis before the patient’s health deteriorates.

This paper will highlight the development and utilization of coronary artery stent for diagnosis. It will also provide an overview of the pressure sensor integration with stent and X-ray. The monitoring of in stent restenosis will be discussed in full detail as well. The paper will also highlight its development as a treatment also, while also providing a brief overview of hyperthermia treatment and drug eluting stent. The paper will discuss the challenges that limit these devices and provide methods to solve them.

## 2. In-Stent Restenosis Monitoring

Several monitoring methods have been examined for early diagnosis of in stent restenosis. Stents can be integrated with a capacitive pressure sensor in order to create a pressure sensitive circuit that possesses wireless sensing capability. Computed tomography angiography (CTA) and X-ray methods were also utilized to analyze the state of stented blood vessels and the areas surrounding them. A brief discussion of these methods will be done in order to describe their performance and potential for achieving clinical relevance.

### 2.1. Wireless Pressure Sensor Monitoring Method

Constant monitoring of blood pressure using minimally invasive devices that can be integrated in the coronary artery can be used as a diagnostic and early warning system for cardiac health.

Wireless sensors have a wide range of uses in biomedical technologies. For the past sixty years, studies on telemetric medical diagnostics functioning in the radio frequency (RF) have been conducted [18,19,20,21]. Most of these devices work by making use of passive inductor–capacitor (LC) resonant circuits with varying resonant frequencies based on various biological or physiological variables of interest. Therefore, the wireless determination of the resonant frequency is able to achieve the required diagnostic information. One can perform this determination by analyzing the impedance of an antenna that is positioned close to the implant inductor via electromagnetic coupling [22,23]. Vascular applications make use of wireless sensors based on micro-electro-mechanical system (MEMS) technology [24,25,26].

This system makes use of a nodically bonded capacitive pressure sensor and a gold electroplated spiral inductor when the sensor size is 2.6 × 1.6 mm. This device functions well when it is implanted near the body’s surface. However, there is still an issue with size [23].

Pressure is monitored using a microchip with a planar thin film inductor integrated into a micro machined capacitive pressure sensor. However, these designs only have a read range of a few centimeters as a result of the limitations in the magnetic coupling of small inductors [27,28,29].

A biodegradable conductor material, as well as biodegradable polymers polycaprolactone and poly-L-lactide, served as dielectric and structural materials. They were used to design wireless pressure sensor with MEMS and served as an inductor with an area size of 1 cm^2^. This was performed under an operation frequency of 50 MHz [30].

A styrene-butadiene-styrene (SBS) was used instead of other more commonly used elastomers such as polyurethane or poly (dimethyl siloxane) (PDMS) for the design pressure sensor. This was then integrated with inductance because of its low loss in the high frequency range when the size area is 1 × 1 × 0.1 m^3^ [31].

A photosensitive SU-8 polymer was utilized for designing the MEMS pressure sensor and then integrated with inductance when the area size is 3.13 × 3.16 mm × 150 µm. The 3D-printer biodegradable polymer was used to fabricate stent under an operating frequency of 200 MHz [32]. The values of pressure ranged from 0 to 230 mmHg, with a sensitivity value of 0.043 MHz/mmHg, and a reader distance of 10 mm.

A micro pressure sensor made with polymer and incorporated with a 3D-printed polymer stent was designed and developed. The SU-8 bonding process served as a way to obtain uniform sensitivity for the pressure sensor value of around 160 KHz/mmHg and an area size of 4 × 4 mm^2^. This served as a method of measuring the pressure from 0–220 mmHg [33].

In 2018, Park et al. formulated and built a wireless pressure sensor with SU-8. This sensor was then integrated with a biocompatible polycaprolactone stent with a sensor size area of 4 × 4 × 0.15 mm^3^. The sensitivity value was at 160 KHz/mmHg and the operation frequency was 179 MHz [34]. Development was done by fabricating a 3D-printed degradable and biocompatible polymer stent, which was then integrated with a wireless pressure sensor created from poly(D-lactide) (PDLA). PDLA decomposes and is absorbed by the body along with the polymer stent. Furthermore, it satisfies the sensitivity 60 KHz/mmHg given the pressure range of 0–220 mmHg and a sensor area size of 4 × 4 × 0.15 mm^3^, using an operation frequency of 148 MHz [35]. A cost-effective thin-film technique for micromachining surfaces was integrated with electronics comprising a round coil-capacitor array made using flexible electronics to meet the greater than 40% transfer efficiency value for a given tissue thickness of 3.5 cm. The proposed design was able to continuously monitor cardiac pressure across the 5–300 torr range [36]. Figure 2 illustrates the kinds of pressure sensors using circular and planar coils as antennae and incorporated with MEMS capacitance.

To lower the extra antenna components for the wireless pressure sensor to decrease space requirements, researchers are choosing to use the stent as an inductive antenna to establish a wireless connection [37]. The proposed use of the metal stent as an RF antenna and to achieve the telemetric integration of the MEMS-based pressure sensor with stents demonstrated experimental evidence of principle.

In 2006, Takahata et al. constructed a 20 mm long antenna using a stainless-steel stent having a 50 µm layer of planar foil and 3.5 mm diameter. Parylene C was selected as the coating substance to produce a thin, consistent, and conformal coating that is inert to chemicals; it is non-conductive and biocompatible to meet the sensitivity requirement of 273 ppm/ton with the space being lower than 1 cm when measured from the stent [38].

In 2009, an antenna design using stents was demonstrated for power transfer and wireless telemetry in implanted electronics applications. A stent of 15 mm length and 5 mm diameter was evaluated in free space, while another stent having 35 mm length and 5 mm diameter was evaluated in vivo. The electronic package had an area of 1 mm^2^ and a thickness of 300 µm and used an operational frequency of 2.4 GHz to meet the 3.5 cm distance requirement for reading the data using a +8 dB gain [39].

In 2009, one more researcher used a 30 mm long stent having a 5–6 mm diameter as an antenna; the stent was settled at 3.5 cm inside the chest to obtain a 32–35 dB gain at a receiving distance of 10 cm [40].

A smart stent having an in-built MEMS pressure sensor was built using medical-grade stainless steel. The stent design had a helix-like pattern to ensure that it not only worked as an inductive antenna but also complies with the commonly used balloon catheter stenting technique. The stent was electroplated with a layer of gold to decrease its series resistance. The stent had a 20 mm length and 5 mm diameter. Its sensitivity in free space was 302–335 ppm/mmHg up to a pressure of 250 mmHg, while the in vitro sensitivity was 146 ppm/mmHg when operated in the frequency range of 30–80 MHz and sensor size is (1.5 × 1.5 mm × 200 µm) [41].

The smart stent was produced with an integrated capacitive MEMS pressure sensor from stainless steel and electroplated with gold to decrease the series resistance of stent; the stent was passivized by a Parylene C layer to make its surface electrically isolated and biocompatible. The stent length was 30 mm and chip sensor size (1.5 × 1.5 mm × 200 µm) by using operation frequency 10 MHz in free space and in vivo, respectively [6,42].

To halt the random presence of electric current and to attain adequate radiation performance, the stent antenna was built using multiple rings having struts and a crown. Each ring uses one connector to establish the connection. The stent was 18 mm in length and consisted of 6 crowns and 9 rings having a 2 mm diameter prior to expansion achieved using 2.3 GHz radiation. The gain was determined to be 1.38 dBi, while the radiation efficiency was 74.5%. The stent was built using an L-605 cobalt-chromium (Co-Cr) alloy because the alloy allowed for lesser thickness compared to stainless steel [43]. The removal or reduction of struts caused the structural integrity to decline because of additional stretching, spiral contraction, or compression, thus causing motion and impacting stent performance. Figure 3 illustrates the kinds of pressure sensors using Stent as antenna and incorporated with MEMS capacitance.

Table A1, showing the performance of the pressure sensor, was done by the preview researcher.

### 2.2. CTA Monitoring Method

For a stent placement inside a coronary artery, computed tomography angiography (CTA) is a non-invasive imaging technique for follow up consultations [45]. Figure 4 depicts the CTA approach.

Invasive coronary angiography (ICA) has been the gold standard for diagnosing in-stent restenosis (ISR) because the coronary vessel lumen can be visualized directly with high resolution, both temporally and spatially. Nevertheless, the invasive nature of the ICA procedure also has a high associated risk of mortality and morbidity. Additionally, ICA is an expensive procedure that requires highly skilled and expert personnel [47,48]. Hence, there is a high clinical relevance to work on a prospective non-invasive, cost-effective, and reliable detection technique [49]. The growth of computing power has allowed iterative reconstruction to be used in the clinical domain, which provides significant advantages in the imaging of the human body [50,51,52]. These encompass enhanced image quality, decrease in image noise, and the prospect for savings in the radiation dose. 

Multi-slice spiral computed tomography (MSCT) has been the domain of critical research, and it is established that the technique may be used in cases of follow up for coronary artery-related conditions [49,53,54,55,56,57,58].

Increased cardiac motion and artifacts caused by stents during the assessment of stent restenosis using 4- or 16-slice CTA limit the value of this imaging technique. The accuracy may be significantly decreasing along with the possibility of 36% of the stents being inaccessible when imaged using a 16-slice CT [47,55].

Coronary artery ISR detection has been determined to have high specificity (88–100%) when the 64-MSCT technique is used. Better temporal and spatial resolution permit a negative predictive value (NPV; 90–100%) [54,59,60,61,62,63,64,65,66]. In comparison to earlier systems, the new generation MSCT overcomes the restrictions that were caused due to low resolution [67,68].

Systems having a 320-MSCT have been demonstrated to have volumetric coverage as high as 16 cm using a single gantry rotation [69]. Such systems capture 320 slices during one rotation. By the use of a volumetric CT approach for data acquisition, the contrast load and the detection time have been reduced for the evaluation of CT coronary angiography (CTCA). These systems offer significant advances in the evaluation of coronary artery disease [70,71,72,73]. Nevertheless, bare-metal stents or drug-eluting stents (DESs) cause blooming due to their metallic parts. Hence, the use of MSCT angiography to evaluate such cases remains limited, especially if the stent size is ≤3 mm [70].

Dual-source computed tomography angiography (DSCTA) equipment uses two X-ray tubes and detectors, which provide for advancement in temporal and spatial resolutions [74,75], thereby facilitating an additional reduction in artifact generation leading to significantly increased quality of cardiac imaging independent of heart rate [76].

Per-segment analysis of coronary artery disease (CAD) attains high SEN (94%) and SPE (97%) when the DSCTA method is employed [77]. Though this technique provides high resolution, both spatial and temporal, along with a marked reduction of the effects of motion and blooming artifacts [78,79], thicker strut slices and higher density stent metal have been more challenging to diagnose [74].

### 2.3. X-ray Monitoring Method

Without a doubt, in the case of coronary artery disease (CAD), if the stents are integrated with X-BP micro-sensors, as depicted in Figure 5, additional follow up and regular monitoring must be performed by medical personnel. Such critical medical requirements need attention, for which a passive, micro-machined X-ray-detectable device is needed for blood pressure measurement [80,81]. Hence, the current invention incorporates measures to handle X-ray monitoring issues:

- To design a pressure-sensing device that can be implanted fully into the body, and is observable using X-rays to determine changes in pressure;

- To provide a pressure-sensing device that can be observed using common and widely available X-ray imaging. Such a device would negate the requirement for advanced monitoring technology or equipment;

- To provide a pressure-sensing device having a compact form, which makes it appropriate for use in areas with little space;

- To provide an implantable pressure-sensing device that may be read using X-rays and that can monitor ventricular pressure and be easily coupled with a ventricular shunt;

- To provide an implantable pressure-sensing device which that may be read using X-rays and can be shaped like a ventricular pressure shunt to provide ease of simultaneous implantation of the device and the shunt as a compact package;

- To provide a passive implantable pressure-sensing device that may be read using X-rays without the need for external power;

- To provide an implantable pressure sensing device that may be read using X-rays and can respond to pressure differential, where it is able to correct the pressure in response to a change in atmospheric pressure.

## 3. In-Stent Restenosis Treatment

The treatment of in-stent restenosis (ISR) is among the significant challenges in interventional cardiology [82]. About 20 to 40 percent of de novo coronary lesions being handled using a bare-metal stent) are affected due to restenosis [83]. Though the process is understood to be benign, there are data that suggest that stent restenosis negatively impacts the long-term survival of patients using coronary stents [84]. Stent restenosis makes it difficult to determine suitable treatment modalities that decrease the risk of recurrence. For ISR, plain balloon angioplasty is considered as the first line of treatment, yet the results have indicated over 40% recurrence, which is disappointing [85]. No additional advantage has been found for alternate interventions like excimer laser angioplasty, rotation atherectomy, cutting balloon, and direction coronary atherectomy [86,87]. There are concerns about its complexity especially considering the extended risk of vessel occlusion [87]. Additionally, the decline in benefits with time [88] causes these techniques to find limited use. For the treatment of ISR, the two successful techniques are discussed briefly.

### 3.1. Hyperthermia Treatment

Hyperthermia treatment involves moderate heating of stents and the approach remains useful in inhibiting cell proliferation without inducing thrombosis [89,90,91]. The primary cause of in-stent restenosis is heating to temperatures near 50 °C [92]. The design principle regarding hyperthermia treatment involves an electro thermally active stent with an electrical inductor function, which operates as a passive inductor–capacitor (LC) resonator when combined with an integrated capacitor. This feature produces heat in an inductive stent, once it is exposed to a radio frequency (RF) electromagnetic field that is tuned to the design resonant frequency particular to the device [93,94,95], as illustrated in Figure 6.

Special heater-equipped catheters that are surgically inserted at the position of the stent [91], involve invasive procedures that constrain implementation of the technique as well as increases in treatment cost.

Higher power levels of some 1–10 KW were applied to initiate inductive heating [99,100], which is neither safe nor practical in clinical applications.

Temperature regulation as used in hyperthermia treatment is mandatory for avoiding the high temperatures that injure healthy tissue. Consequently, current design trends explore the integration of multifunctional systems.

MEMS transducers and other microsystems offer a practical path to realizing electro thermally active stents with temperature regulation and controls, which enable their reliable and safe use in endohyperthermia treatments of restenosis [44,101].

Shielded wireless hyperthermia treatments were designed with wireless micro heaters [97] which contain circuit breakers that work to regulate implant temperature and are readily integrated in stents for deployment within a blood vessel.

Wireless power transfer serves a key function in enabling active stent applications for hyperthermia therapies of in-stent restenosis. Improvements in wireless power transfer are available for increasing Q factors of stents and for efficient power delivery, as well as improved heating efficiency (HE) [96,102].

Table A2, showing the design performance of hyperthermia treatment design, was done by researchers.

### 3.2. Drug-Eluting Stent Treatment

Drug-eluting stents have evolved as the most effective and safest approach in the primary inhibition of restenosis [103,104]. Observational studies have shown promising results in the use of drug-eluting stents for ISR lesions [105,106]. A drug-eluting stent is a bare-metal stent with a drug coating [107] that minimizes neointimal hyperplasia and lessens repeat revascularization in BMS use. DES was developed through the application of different types and combinations of drugs as well as design materials and structural schemes.

To avoid smooth cell proliferation and lessen restenosis in comparison to BMS, first-generation versions of DES were developed using paclitaxel and sirolimus drugs [108,109,110]. Sirolimus drugs offer greater antirestenotic efficacy than similar paclitaxel drugs, in terms of better kinetics and a broader therapeutic index [111,112]. This particular generation features very late thrombosis, low restenosis, and adequate mechanical performance.

The limus family of drugs, which include zotarolimus [113], everolimus [114,115], umirolimus [116], and amphilimus, were used in the development of second-generation DES designs [117]. These feature specialized properties, i.e., zotarolimus provides a highly lipophilic analogue of sirolimus, everolimus provides a much increased interaction with the mechanistic targets of rapamycin complex 2, while umirolimus features lipophilic properties that are about 10 times greater than that for sirolimus.

Third-generation DES utilized biolimus [118] and also novolimus [119], an active metabolite variant of sirolimus that is shown to be a potent inhibitor of smooth muscle cells in various in vitro studies. Second-generation as well third-generation designs feature lower restenosis, lowered rate thrombosis, and higher mechanical performance.

Fourth-generation DES comprise bioresorbable stent designs [120]. Bioresorbable stents avert acute vessel closure and also recoil by transient scaffolds at the vessel. Furthermore, these are fully biodegradable scaffolds that elute antiproliferative drugs, which inhibit neointimal hyperplasia and constructive remodeling [118]. This generation features very late and very early thrombosis, acceptable restenosis, and less mechanical performance.

Different materials were utilized in varied generations of DES to suspend in-stent restenosis. Primary design principles resort to the use of polymer materials that behave predictably and deliver capably in time and dosage, feature low inflammation reaction, are highly elastic, and do not modify drug activity or affect the structure of the device [121].

The durable polymers utilized in the first-generation versions of DES stimulated constant arterial wall inflammation that delayed vascular healing, which furthered stent thrombosis while delaying in-stent restenosis [122].

Biodegradable polymers as utilized in newer generation designs present lower risks of late thrombosis than the durable polymers of the first-generation versions. Stents made of biodegradable polymers require briefer double antiplatelet therapy than those with durable polymers [123].

Polymer coating plays a key part in DES in terms of constraining smooth cell propagation, decreasing drug dose as well as polymer exposure, diminishing platelet adhesion and thrombus creation, improving the advancement of endothelial cells, and acting like an accelerator of endothelialization such as abluminal coating [124], vinylidene-fluoride hexafluoropropylene copolymer [125], lactic or glycolic acids [126], silicon carbide [127], carbofilm [128], and titanium nitride oxide [129].

DES design also has a key function in preventing neointima proliferation. No surface coating is required when a channel or depot stent (reservoir) is used, which may be loaded with a single, or else multiple, drugs for programmed delivery [107,130,131].

Stent performance can be improved through increased strut thickness, which enhances radial strength, radio visibility, as well as arterial wall support. Conversely, this can lead to more vascular injuries and trigger further intimal hyperplasia, leading to higher risk of restenosis than that with thinner strut designs [132,133]. In efforts to decrease strut thickness further while maintaining sufficient radio visibility as well as radial strength, new metallic materials including cobalt chromium alloys are being developed in the manufacture of stents [134,135].

## 4. Current Challenges and Problems

The bio implantable devices technology has been recently used for medical applications in the human body by monitoring or recording many vital activities such as blood flow in the arteries and the pressure sensors placed on them, as well as monitoring the state of the in-stent implanted in the artery and the in-stent restenosis over time. The in-stent geometry such as shape, size, and design, and the coating material are still the biggest challenges. Hence, it is necessary to choose carefully among coating materials of suitable thickness and quality. To make this manuscript more readable to the readers, and based on what was introduced in this review in Table A1 and Table A2, the most important challenges for the in-stent implants and pressure sensors can be summarized as follows.

### 4.1. Pressure Sensor Properties

Pressure sensors should be made increasingly accurate and sensitive, and have more resolution. These aspects depend on the variety of materials used to design and produce the sensors as well as the coatings applied to them. Accuracy enhancement comprises several basic aspects such as enhanced isolation and blocking ability from media capable of causing external interference, applying a dense pin-hole coating on the pressure sensor surface to prevent loosening, which may cause errors. Lightweight and soft construction are beneficial since a more massive device restricts the flexibility of the dynamic pressure sensor structure. Therefore, a coating substance having less weight and Young’s modulus is required. Additionally, the lightweight aspect is also beneficial since it can fill defects like surface clearance. Said differently, lesser coating thickness enhances sensitivity, while resonant peaks lessen with decreasing electronic insulation. Contrarily, an increase in coating thickness causes lesser sensitivity, and resonant peaks rise with an increase in electronic insulation. Parylene C, PDMS, and silicon having a 110-crystal orientation are probable substances that may be used to produce sensor diaphragms needing large deflection along with high biocompatibility and sensing ability.

### 4.2. Material Consideration

Thinner struts are preferred in new stent designs, for these benefit more from further reductions in clinical and angiographic restenosis than variants with thicker struts. Furthermore, fewer strut numbers present a lower risk of restenosis when compared to more struts. Moreover, thinning struts may lessen their structural integrity through spiral contraction and new stretching motions that degrade performance. It is therefore necessary to choose suitable materials to resolve the problem, such as magnesium, cobalt chromium alloy, and innovate materials to overcome the current issues.

### 4.3. Stent Geometry

The distance between the implanted stent and the reader coil is determined by data transfer and power transfer efficiency requirements for reading pressure sensor data, as well as efficient heating (EH) criteria for hyperthermia treatment. Increases in stent lengths not exceeding 40 mm leads to increased power transfer efficiencies and mutual inductance, based on the ISO 25539 commercial stent standard. However, increases in length leads to increases in the tissue surrounding the stent and inhibits problematic restenosis. Strut cross-sectional area is also critical in-stent restenosis. A square strut cross-sectional area with sharp edges is not recommended, for it will interfere with fluid blood flow and may also slice blood cells. Round strut cross-sectional areas without corners and sharp edges are safer in reducing restenosis.

### 4.4. Sensing Distance and Range

Sensor range needs to be appreciably increased in order to attain clinical relevance, which is satisfied with mmHg and reader distances exceeding 35 mm. This difficulty must be resolved through further improvements in stent implant design, device quality, external reader antenna, and external reader alignment with implanted antenna. Wireless transmission of data and power can be achieved effectively using inductive coupling, where 35 mm of separation is required for transferring power from the outer part to the inner. Typically, radio frequency techniques transfer a short pulse of low power using the coil of the reader antenna and establish a fixed amplitude sinusoidal carrier system for steady wireless power transmission. The system uses a component of the primary coil and deploys it as a transmitting antenna, which is placed externally. In contrast, the secondary coil is internal and behaves as a receiver.

### 4.5. Miniaturize Integrated Circuit

Typically, the stent implanted inside should have minimum dimensions so that it is less invasive and less prone to in-stent restenosis of the coronary artery. Reduction in the dimensions of implantable devices may be achieved using auto-zeroing methods and artificial intelligence. MEMS capacitive sensors and circuit breakers having reduced dimensions allow for the reduction in chip size, thereby reducing the need for space, facilitating convenient stent expansion, a less invasive stent placement procedure, and cost-effectiveness. Additionally, there is a possibility to use shapes other than the typically used rectangular shape to prevent sharp edges from causing injury.

### 4.6. Monitoring Methods

Certainly, the CAD and integrating X-BP micro-sensors into a stent required follow up and occasional monitoring by the medical staff, and to address these insistent medical needs a micro-machined, passive, X-ray-detectable blood pressure sensor needs to be developed. Therefore, there are several monitoring challenges for X-rays, such as changing the pressure through the use of X-rays and being able to conform to the shape of a ventricular pressure shunt to facilitate the implantation of the pressure sensor and the shunt simultaneously. However, a passive X-ray readable is required which does not need any power.

## 5. Conclusions

According to the approved medical statistics, it has been observed that restructuring the stent embedded within the coronary artery post re-opening through the catheterization procedure leads to clotting of blood, thereby raising the risk. Hence, for this study, the stenosis development history concerning cardiac stents and techniques for early diagnosis and treatment was evaluated. A concise survey of many research works pertaining to quick detection and treatment was carried out, as depicted in Table A1 and Table A2.

Through this review, it was determined that early detection techniques need further development, including pressure sensors, CTA, and X-ray. Specific to the coronary artery, the pressure sensor technique is prone to inaccuracy and sensitivity issues. Sensor material and shape play a significant role in determining how frequently these issues arise. The review provides evidence that the substances preferred to build these sensors are Parylene C, PDMS, and silicon with a crystal orientation of 110. Concerning the sensor size and shape, an edge-free circular geometry is preferred since it prevents tissue growth, which is responsible for restenosis. In contrast, a reduction in the sensor dimensions provides ample space for blood flow while reducing the incidence of tissue regrowth. CTA is another early restenosis-detection technique. It has been observed that CTA has issues concerning precision and resolution specific to ascertaining stenosis rate, especially for arteries with a diameter of 3 mm or less.

Additionally, there are several techniques like drug-eluting stents and hyperthermia for managing restenosis. The data obtained from the survey made it evident that hyperthermia has superior results compared to the other techniques since it leads to enhanced stent life and also prevents tissue growth, thereby reducing the incidence of restenosis and chances of thrombosis. However, there needs to be an improvement in the heating efficiency in the coil placed inside the artery. Another aspect that demands attention is less efficiency during wireless power transmission, which is the result of the significant distance between the outer and inner parts. Hence, we believe that the stent may be used as an antenna coil and may replace supplemental circular or planar coils for receiving power. The stent shape and dimensions may be modified to allow them to receive energy and also provide longer life, thereby having a longer-term treatment. This study is expected to introduce a key database of associated works for researchers in the field. Lastly, the problems and challenges involved in improving the devices and long-term treatments were discussed in detail.

## Figures and Tables

**Figure 1 sensors-20-04303-f001:**
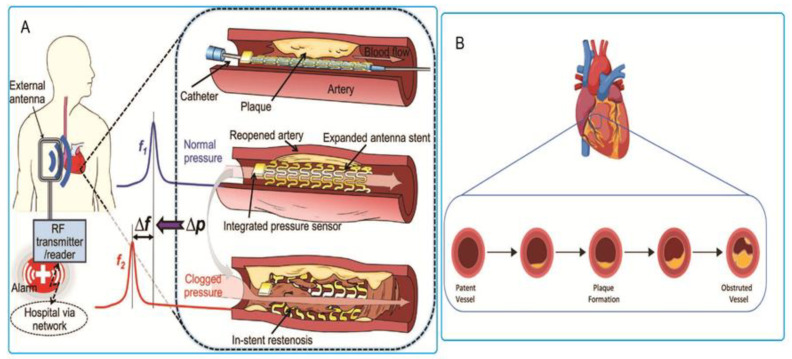
(**A**) Procedure of percutaneous coronary intervention (PCI) or coronary angioplasty. Reproduced with permission [6], Wiley’s Open Access; (**B**) elastic recoil and negative remodeling contribute to stenosis. Reproduced with permission [7], Wiley’s Open Access.

**Figure 2 sensors-20-04303-f002:**
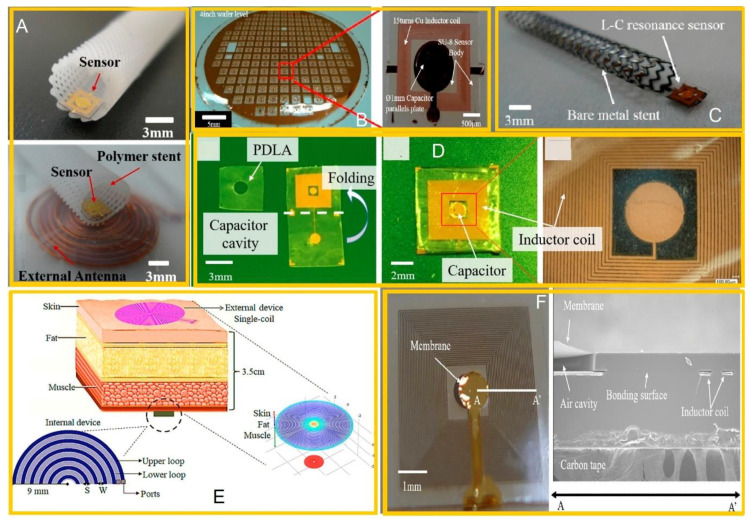
Wireless pressure sensor in micro-electro-mechanical system (MEMS) device: (**A**) Optical images of polymer stents integrated with wireless pressure sensors; (**B**) actual photo images; (**C**) optical image of bare metal stent integrated with the wireless pressure sensor. Reproduced with permission [32], Sensors Open Access; (**D**) fabricated poly(D-lactide) (PDLA)-based wireless pressure sensors. Reproduced with permission [35], copyright 2020, Elsevier; (**E**) scheme showing the inductive coupling across the biological tissue. Reproduced with permission [36], Sensors Open Access; (**F**) fabricated wireless pressure sensor with SEM cross sectional view of the wireless pressure sensor fabricated using a SU-8 thermal pressure bonding technique. Reproduced with permission [34], copyright 2020, Elsevier.

**Figure 3 sensors-20-04303-f003:**
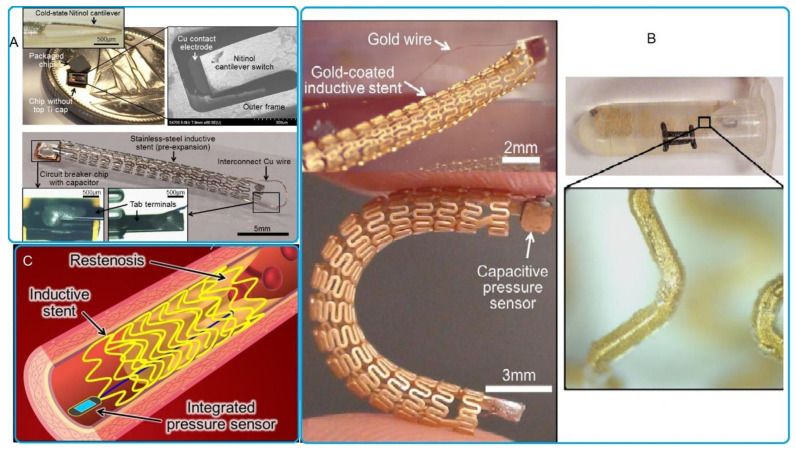
The pressure sensor using a stent as an antenna integrated with MEMS capacitive: (**A**) Completed active stent device with the stent terminal on which the interconnect wire was bonded. Reproduced with permission [44], copyright 2020, IEEE; (**B**) fabricated LC-tank stent devices integrated with capacitive pressure sensors and a device coated with Parylene C; (**C**) conceptual diagram of wireless sensing of vascular conditions through the sensor-integrated smart stent. Reproduced with permission [41], copyright 2020, Springer Nature.

**Figure 4 sensors-20-04303-f004:**
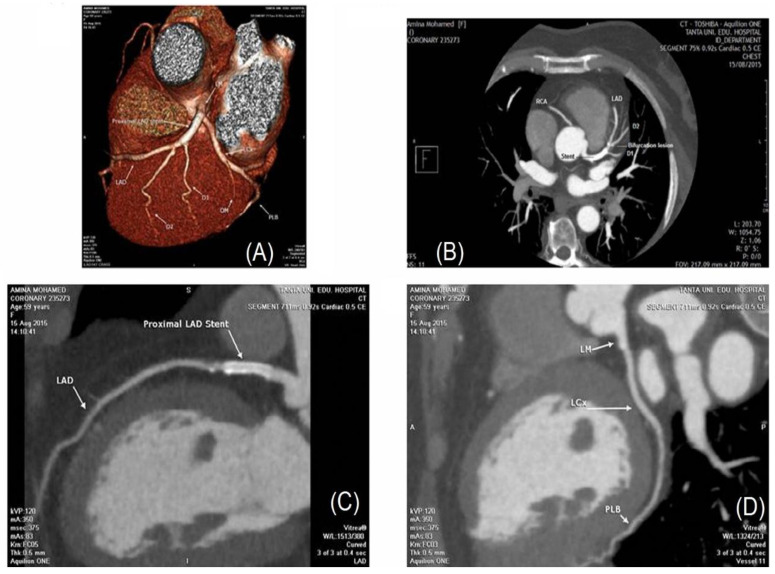
Computed tomographic (CT) angiography: (**A**) Left anterior oblique view of 3D volume rendered image showing the position of the proximal LAD stent; (**B**) Axial MIP image showing the position of proximal LAD stent; (**C**) Sagittal curved planar reformatted image showing patent proximal LAD stent; (**D**) Coronal curved planar reformatted image showing patent dominant LCx and PLB. Reproduced with permission [46], copyright 2020, Elsevier.

**Figure 5 sensors-20-04303-f005:**
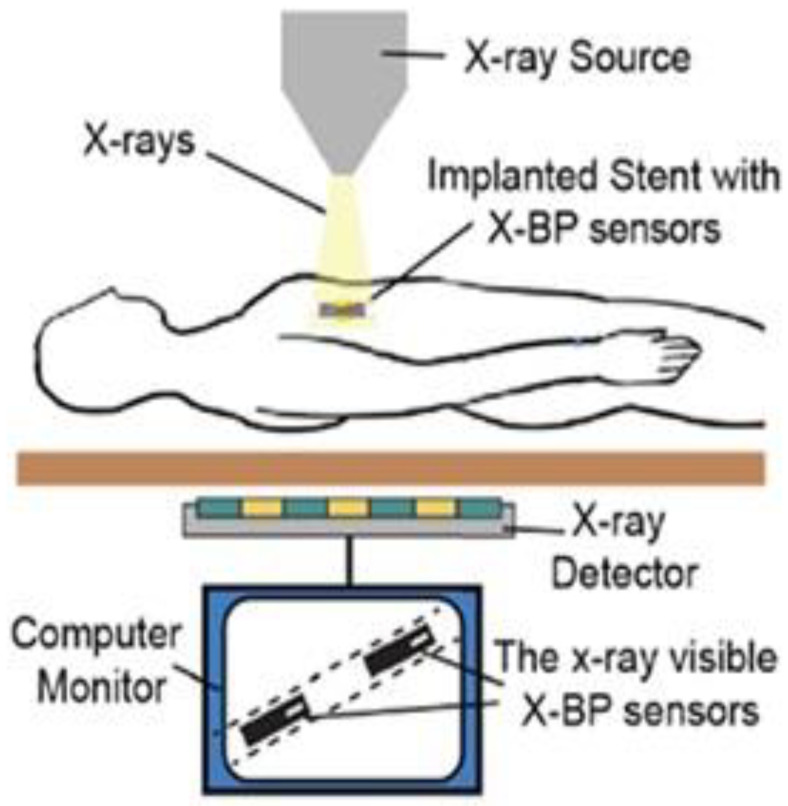
X-ray-based pressure-sensing technology for monitoring restenosis non-invasively. Reproduced with permission [81], copyright 2020, IEEE.

**Figure 6 sensors-20-04303-f006:**
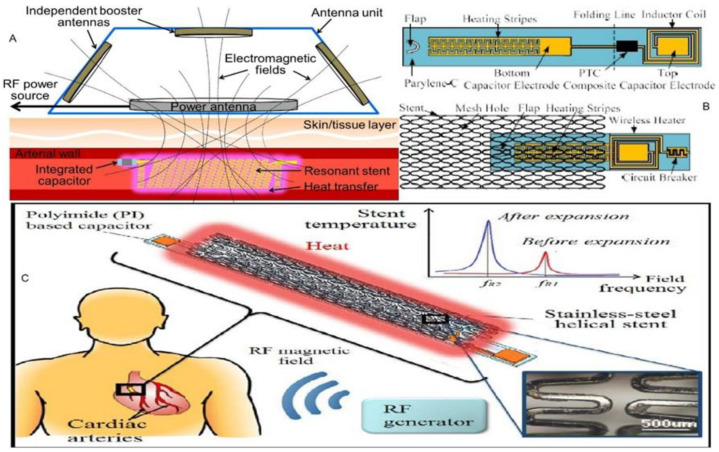
Hyperthermia treatment: (**A**) Conceptual diagram of the stent-based endohyperthermia system using the resonant-heating stent powered and operated using the external RF antenna unit with independent booster antennas. Reproduced with permission [96], copyright 2020, IEEE; (**B**) schematic of the proposed wireless micro heater. Reproduced with permission [97], copyright 2020, IEEE; (**C**) conceptual illustration of the stent-based wireless endohyperthermia for in-stent restenosis treatment. Reproduced with permission [98], copyright 2020, IEEE.

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
