# Peer review of "Health Care Monitoring and Treatment for Coronary Artery Diseases: Challenges and Issues"

_sensors, 2020, doi:10.3390/s20154303_

Round 1

Reviewer 1 Report

Dear authors,

In section 2.1, You presented a brief of each wireless pressure sensor in figure 2, except for reference [36]. In order to keep the same format, it is important to write a brief about reference [36], as the other references in figure 2.

In line 370, I think that correct sentence is: "The bio implantable devices technology...."

In section 4.1, you mention about pressure sensor properties, in specific the thickness and quality of coating material. However, you do not mention the real impact of select the best suitable cover material: biocompatibility and the effect of the thickness of this coating material over the mechanical diaphragm in the operating state of pressure sensors.  It is very important include this effects in the paragraph, because this is a review, and in consequence you will show the actual challenges in this devices.

In general, this article is a great job an present a good aportation; however, for scientists and specialist in the field, that commonly read a paper review for identifying the main ideas about an specific theme, it is necessary that conclusions could be improved to reader can identify the main challenges and problems in a specific aspect, because now, the conclusions are written in a general way.

Reviewer 2 Report

The authors have come up with a comprehensive review on the state-of-the-art in stent monitoring, ending with the challenges and issues laying ahead. The review was nicely written, but I would like to make a few recommendations and suggestions.

1) I wish the title would be refined. At first glance, it seems to be towards health care monitoring in general for coronary artery diseases, which could cause the reader to imagine the extent of the review to include detection, post-treatment and prognosis.

2) The first line of abstract is grammatically inaccurate. 

3) The authors start off with the state-of-the-art on wireless monitoring using passive circuits, but continued in more traditional methods such as imaging. I wish there would be a better flow into each methods, as it seems rather compartmentalized.

4) The challenges and issues should be better written to address the advantages, flaws and cons of each method, and if possible, the future prediction as well.

5) I am wondering if the authors could cover active circuits that has been developed for in-stent restenosis monitoring as well. 
